# Protein Evaluation of Feedstuffs for Horses

**DOI:** 10.3390/ani13162624

**Published:** 2023-08-14

**Authors:** Franziska Bockisch, Johannes Taubert, Manfred Coenen, Ingrid Vervuert

**Affiliations:** Institute of Animal Nutrition, Nutrition Diseases and Dietetics, Faculty of Veterinary Medicine, University of Leipzig, An den Tierkliniken 9, 04103 Leipzig, Germany; franziska.bockisch@gmail.com (F.B.); hannes.taubert@uni-leipzig.de (J.T.); manfredcoenen@web.de (M.C.)

**Keywords:** amino acids, digestibility, feed, horse, lysine, protein evaluation system, neutral detergent fiber-bound crude protein

## Abstract

**Simple Summary:**

To formulate rations for horses, feeds must be evaluated for their supplies of nutrients. For this purpose, the German Society of Nutrition Physiology (GfE) proposed a new protein evaluation system that based the protein evaluation of feeds on predicted digestibility in the small intestine depending on chemical properties. A total of 71 feeds for horses were chemically tested and evaluated according to the new protein evaluation system. In a feeding trial, four feeds were randomly tested for post-prandial plasma lysine responses to determine the lysine supplied by the respective feed in horses. Chemical-based feed evaluation confirmed protein-rich feeds as protein suppliers with high amounts of small intestine-digestible protein. Feeding with synthetic amino acids, including lysine, induced a high post-prandial plasma lysine response in horses. Although limited by a low number of horses, high plasma lysine responses after feeding lucerne seemed not to correspond to the chemical feed evaluation, as chemical protein evaluation indicated low pre-cecal protein availability of lucerne. Protein evaluation by chemical parameters seemed to be limited, at least for forages such as alfalfa, as the impact of chewing and bacterial fermentation processes were neglected. Post-prandial plasma lysine levels may provide information regarding the chemical protein evaluation of feeds.

**Abstract:**

The German Society of Nutrition Physiology has proposed a new protein evaluation system for horse feeds to estimate pre-cecally digestible crude protein (pcdCP) and amino acids (pcdAA) from chemical properties. A total of 71 feeds for horses were chemically tested and evaluated according to the new protein evaluation system. A feeding trial with eight horses tested whether differences in estimated pcdAA and neutral detergent soluble CP (NDSCP) in the diet were reflected by post-prandial (ppr) kinetics of plasma lysine (Lys) by feeding a complementary feed (control = CTRL) with 1.02 g Lys/100 kg body weight (BW) as well as three diets with 3.02 g Lys/100 kg BW, as follows: (i) CTRL with synthetic AA (CTRL + synAA); (ii) CTRL with soybean meal (CTRL + SBM); and (iii) lucerne pellets (LUC). In comparison to CTRL, the areas of curves (*AUC*s) of ppr plasma Lys differed: CTRL < CTRL + SBM (*p* < 0.01) < CTRL + synAA (*p* < 0.05). For 71 feeds, the estimated pcdCP was correlated with the CP content (*p* < 0.001), NDSCP (*p* < 0.001), and ash-free neutral detergent fiber (*p* < 0.001). A mean neutral detergent insoluble CP content of at least 3–5% can be assumed in horse feed. It is speculated that the predicted availability of Lys from LUC seems to be underestimated by the new protein evaluating system. The influence of chewing and microbiota in vivo needs to be considered in horses.

## 1. Introduction

Since 2014, the new equine German protein evaluation system has been used to estimate the extent of auto-enzymatic digestible protein that can be absorbed from the small intestines of horses. For the new system, the German Society of Nutrition Physiology (GfE) [1] assessed the data reported in the literature and concluded that pre-cecal amino acid (AA) absorption occurs only in the small intestine. In contrast, post-ileal AA absorption from microbial protein production or protein breakdown in the hindgut does not contribute to the AA supply of horses [2,3,4,5,6,7,8,9,10]. Subsequently, the recommendations for CP and AA supply in horses should be formulated based on pre-cecal digestible protein (pcdCP) and pre-cecal digestible AA (pcdAA). Therefore, to evaluate protein in horse feeds, CP is assigned to two fractions using solubility according to the Cornell Net Carbohydrate and Protein Evaluation System in cattle: neutral detergent insoluble CP (NDICP) and neutral detergent soluble CP (NDSCP) [11,12]. A portion of CP is associated with neutral detergent fiber (NDF) and can be analyzed in the NDF fraction as insoluble CP (NDICP). The analytical parameter of the cell-wall-bound fraction (NDICP) represents the pre-cecally undigestible portion, which can be used to estimate the soluble fraction of protein, named NDSCP (neutral detergent soluble CP), and to determine the difference between CP and NDICP. Subsequently, the calculated non-cell-wall-bound protein (NDSCP) represents the potential pre-cecally digestible protein fraction. The NDSCP is assumed to be pre-cecally digestible at 90%. Consequently, the pcdCP can be estimated [1]. The AA profiles of NDICP and NDSCP appear to be similar regardless of their solubility [13,14,15,16,17,18,19,20]. For AA, like for the protein, it is assumed that 90% of the NDSCP-bound fraction is pre-cecally digestible (pcdAA). In that context, a correlation between the Lys content of the diet and post-prandial (ppr) plasma Lys levels is an established parameter in swine [21]. Furthermore, in horses, there is a dose-dependent post-prandial peak plasma AA response within 180 min [15,22,23,24]. 

The present study aimed to test 71 typical feedstuffs for horses according to the new protein evaluation system. We hypothesized for the new protein evaluation system that a higher amount of CP would be related to a higher estimated pcdCP content in feeds. The following feedstuffs were included: roughages, grains, legumes, seeds, and compound feeds for horses were analyzed to calculate protein fractions.

In addition, to evaluate pcdAA availability from feedstuffs in horses, a feeding trial was performed to investigate whether ppr changes in AA (e.g., Lys) reflect the analytical protein evaluation system for different feedstuffs. We hypothesized that diets with equal Lys levels but different NDICP contents would affect ppr Lys responses.

## 2. Materials and Methods

### 2.1. Analytical Protein Evaluation

#### 2.1.1. Sample Collection

In total, 71 feedstuffs for horses were included in the current study. The feeds covered the spectrum of typical feeds for equines, including the most important feed groups, such as roughage, cereals, legumes, high-fat seeds, and complementary feeds for horses with specific requirements, such as breeding, sport, and special breeds.

Representative samples were taken according to the standard protocols of the Association of German Agricultural Analytic and Research Institutes VDLUFA [25]. The feedstuffs were categorized into six groups with respect to their botanical origin, their CP levels, and ash-free neutral detergent fiber (aNDFom) (Table 1).

The roughages included hay (*n =* 7), artificially dried forages (*n =* 5), lucerne products (*n =* 5), maize plant cobs (*n =* 1), and alkaline-treated straw samples (*n =* 1). A separate grass group contained fresh grass samples (*n =* 11, first cut). Grains such as oats (*n =* 2), rice (*n =* 3), corn (*n =* 2), spelt (*n =* 1), wheat (*n =* 1), and barley (*n =* 2) were included in the grain group (*n =* 11). Pea flakes (*n =* 1), soybean products (*n =* 5), and linseed products (*n =* 3) were pooled in the legume and seed group (*n =* 9). Compound feeds were represented by six products for breeding horses and five for performance horses. The remaining feedstuffs, including brewer’s yeast, beet pulp, banana chips, compound feeds for special breeds and old horses, and two supplements with synthetic amino acids (synAA), were grouped as “Others” (*n =* 10).

#### 2.1.2. Crude Nutrients in Feedstuffs

Feed samples were ground to 1 mm in size. The dry matter (DM) content was determined after oven-drying (103 °C) to a constant mass. After oven-drying, samples were immediately stored in a desiccator until further analysis. Crude nutrients (CP; CL, crude lipid; CF, crude fiber; CA, crude ash) were determined by Weende analysis [27]. Ash-free neutral detergent fiber (aNDFom), acid detergent fiber (ADFom), and acid detergent lignin (ADLom) were analyzed according to the method of van Soest et al. [28], with minor modifications with respect to the Fibertec^®^ 8000 equipment (Tectator, Rellingen, Germany). The neutral detergent insoluble crude protein (NDICP) was determined by NDF-analysis with subsequent nitrogen (N) determination of the entire filtration residue [28]. This method was modified according to the ANKOM Technology filter bag technique, using fiber bags (F57, ANKOM Technology, Fairport, New York, NY, USA) with pore sizes of 25 microns. Samples with >5% CL content were pre-treated by soaking in acetone for 12 h. The N-amount of the filter bag residue was analyzed using the Kjeldahl method (KjelROC Analyser, Opsis Liquidline, Furulund, Sweden) with minor modifications using 12 mL of concentrated sulfuric acid and two Kjeldahl tablets (1.5 g potassium sulfate, 0.0075 g selenium) to dissolve the F57 bag material. The amount of N in the weighed NDF residue multiplied by 6.25 corresponds to the CP amount that is bound to NDF (NDICP). All feed samples were analyzed in duplicate or triplicate according to the official VDLUFA recommendations [27]. The variation in the analyses for the different parameters of feedstuffs always resulted in findings below the required official upper benchmarks. The crude nutrient contents of the feedstuff groups are listed in Table 1.

### 2.2. Feeding Trial

Four feedstuffs ((i) CTRL; commercial muesli for sport horses; (ii) synAA, supplemented with synthetic amino acids; (iii) SBM, soybean meal (solvent extracted) and (iv) LUC, lucerne pellets) were included in the feeding trial for the new protein evaluation system. The CRTL diet led to an intake of 1.02 g/Lys per 100 kg body weight (BW; as measured on a digital scale at the beginning of the feeding trial as well as the day before blood collection). With respect to the different NDICP contents of the feedstuffs, the three test diets (CTRL + synAA, CTRL + SBM, and LUC) were standardized to 3.02 g/Lys intake per 100 kg BW. 

#### 2.2.1. Animals

Eight Warmblood geldings (mean age ± SD, 19 ± 1 y; mean BW ± SD, 622 ± 27 kg) owned by the Institute of Animal Nutrition, Nutrition Diseases & Dietetics, Leipzig University, were included in this study. All horses were clinically healthy and regularly vaccinated against tetanus. Ten days prior to the beginning of the feeding trial, all horses were dewormed with ivermectin (Eraquell^®^, Virbac Tierarzneimittel GmbH, Bad Oldesloe, Germany). All equines were housed in individual box stalls with straw as the bedding material. Horses were turned out daily onto a dry lot for approximately 5 h/day, except on blood sampling days. Horses were provided with *ad libitum* access to tap water.

The project was approved by the Ethics Committee for Animal Rights Protection of the Leipzig District Government (TVV 18/15), in accordance with the German legislation for animal rights and welfare.

#### 2.2.2. Feedstuffs

The ingredients of the CTRL were 40% flaked maize, 40% flaked barley, 5% rapeseed oil, 4% wheat semolina bran, 2% wheat bran middling, 2% dried, molassed sugar beet pulp, 2% sunflower hulls, 1% maize gluten feed, 0.5% brewer’s yeast, and 3.5% vitamin/trace mineral premix. The synAA supplement (pelleted) was composed of 17% wheat bran middlings, 16% semolina wheat bran, 13% skimmed milk powder, 10% maize middlings, 4% refined rapeseed oil, 3% beet molasses, and 1% fenugreek, as well as fortified with 9% L-lysine hydrochloride (composition according to manufacturer specification). The LUC feed was a warm air-dried material from the complete plant that was pressed into pellets with diameters of 10 mm. The chemical composition is shown in Table A2.

#### 2.2.3. Lys in the Protein Fractions

To determine Lys by means of high-performance liquid chromatography (HPLC), 250 mg of the non-ashed NDF fraction was hydrolyzed in a laboratory bottle by adding 50 mL of hydrolysis solution (1 L consisted of 350 mL double distilled water (H_2_O_bidist_), 492 mL 37% hydrochloric acid, 1 g phenol, made up to 1 L with water). The laboratory bottles were incubated at 110 ± 1 °C for 1 h with the lid off. Subsequently, the bottles were closed and incubated overnight. An aliquot of 1.5 mL (cooled to 20 °C) was centrifuged (6595× *g*, 4 °C) for 10 min. Pre-column derivatization was achieved according to the method described by Ebert [29], including small modifications. The derivatization mixture was prepared in a ratio of 7:1:1:1 (96% ethanol: H_2_O_tri_: 99.5% trimethylamine: 100% phenyl isothiocyanate), and 20 µL was added to the solvent-evaporated sample or the standard mixture. The reaction mixture was incubated for 20 min in the dark, followed by a vaporization step via lyophilization. The samples and standards were redissolved in 200 µL of solvent A (20 mM potassium phosphate, pH 6.8, 5% acetonitrile) and filtered through a 0.2 µm membrane. An AA standard solution (Sigma-Aldrich, St. Louis, MO, USA) for calibration was treated in the same way. The final concentration of the internal standard (norleucine) and the AA standard solution was 25 µM. Lys determination was performed by HPLC as phenylthiocarbamyl derivatives using a reversed phase column (Hypersil, 5 µM, C18, Altmann Analytic, Munich, Germany) tempered at 44 °C. The AA derivatives were separated using mobile phase solvent B consisting of 60% acetonitrile at a flow rate of 1.0 mL/min. The linear gradient elution program was configured to 40.5% solvent B in 40.5 min, ramped to 82% (44 min) and 100% (44.5 min) of B, and held until the end of the run (50 min). The analytes were detected by measuring absorbance at 254 nm. For total lysine determination, ground and dried feedstuffs were used directly for hydrolysis.

#### 2.2.4. Diets and Experimental Design

The horses were fed a basal diet of 1.7 kg meadow hay/100 kg BW that met or exceeded the maintenance requirements for ME and pcdCP according to the GfE [1] (ME intake: 11.2 MJ ME/100 kg BW; CP intake: 173 g CP/100 kg BW). The hay diet was provided via two meals per day. Horses were assigned to the four dietary treatment groups. All test diets were fed in a 4 × 4 Latin square design, as follows:

As a control, 0.3 kg CTRL/100 kg BW with 1.02 g Lys per 100 kg BW was fed to the horses. Furthermore, to reach a target Lys level of 3.02 g per 100 kg BW, the following three test diets were designed:(1)CTRL + SBM: 0.3 kg CTRL/100 kg BW and 70 g SBM/100 kg BW;(2)CTRL + synAA: 0.3 kg CTRL/100 kg BW and 31 g synAA/100 kg BW;(3)LUC: 0.45 kg LUC/100 kg BW.

Except for the CRTL diet, the test diets were equal in Lys levels but different in calculated pre-cecal digestible protein levels. Lys intake was calculated by excluding hay. The horses were adapted to the diets for at least six days. During the adaptation period, the respective diets were fed to the horses once daily in the morning prior to hay feeding. The meal size and chemical composition of the test diets are listed in Table A3.

#### 2.2.5. Blood Sampling Day

Hay and bedding materials were completely removed 8 h before blood sampling. On the sampling day, no hay was fed during the blood collection period. At 0730 h, an indwelling catheter (1.8 mm × 2.35 mm/12 G, B. Braun Vet Care GmbH, Tuttlingen, Germany) was inserted into the *Vena jugularis externa*. The catheter was flushed with physiological saline after every blood sampling. 

#### 2.2.6. Sample Preparation and Chromatographic Analysis of Post-Prandial Plasma Lys

Each blood sample (9 mL) was collected into a tube containing ethylenediaminetetraacetic acid (EDTA, Monovette^®^ 9 mL K3E, Sarstedt AG&Co, Nümbrecht, Germany). 

All sample tubes were centrifuged at 1300× *g* for 10 min, and plasma was stored at −20 °C until further analysis. The first blood sample was collected immediately before feeding the respective test meal at 08:00 h following an 8 h overnight fast. All horses were allowed to consume the test meal for a maximum of 90 min. After the allotted 90 min window to consume the feed, all refusals were removed and weighed. Blood samples were collected every 30 min during the first 4 h, followed by 60 min intervals for the next 4 h. Tap water was withheld for the first 6 h of the collection period.

The post-prandial plasma Lys levels were analyzed similarly to the feedstuff samples using HPLC with pre-column derivatization, with minor modifications. Plasma samples were treated with molecular weight cut-off columns to remove large interfering macromolecules (Vivaspin 500, 3000 MWCO, Sartorius Stedim, Goettingen, Germany). Plasma (500 µL) was centrifuged at 6595× *g* for approximately 30 min at 4 °C, and the flow-through was used for derivatization. Sample concentration was achieved via lyophilization of 100 µL flow-through and 30 µL internal standard (0.25 mM norleucine) after ultrasound treatment. An AA standard solution (Sigma-Aldrich, St. Louis, MO, USA) for calibration was treated similarly. Pre-column derivatization was achieved as described elsewhere, with small modifications [29]. The derivatization mixture was prepared as described for Lys in the protein fractions. Samples and standards were redissolved in solvent A (50 mM sodium phosphate, pH 6.8) and filtered through a 0.2 µm membrane. Lys determination was performed by HPLC, as described for Lys in the protein fractions.

### 2.3. Calculations

Neutral detergent soluble crude protein (NDSCP) was calculated according to GfE [1] and VDLUFA [30]:NDSCP = CP − NDICP   (all values in g kg^−1^ DM)(1)

To calculate the content of pre-cecal digestible protein (pcdCP) and Lys (pcdLys), a digestibility of 90% was assumed according to the literature data [1]. Thus, pcdCP could be estimated using the following equation:pcdCP = 0.9 × NDSCP   (all values in g kg^−1^ DM)(2)

Because the AA profiles of NDSCP and NDICP were assumed to be equal [1], pcdLys could be estimated as [26]:pcdLys = 0.9 × Lys_(NDSCP)_   (all values in g kg^−1^ DM).(3)

The GfE recommendations assume that free AAs are completely absorbed in the proximal small intestine. Thus, in the case of synthetic Lys addition to feedstuffs, an absorption of 100% for the supplemented part was assumed and a conversion factor of 1 instead of 0.9 was used for the calculation. The pre-cecal digestibility for protein and Lys was calculated as:pcDCP [%] = pcdCP × 100/CP(4)
pcDLys [%] = pcdLys × 100/Lys(5)

To determine Lys distribution over the protein fractions, the total lysine content and Lys in the NDICP fraction were analyzed. Subsequently, Lys in the NDSCP fraction was calculated as:Lys_(NDSCP, measured)_ = Lys_(total)_ − Lys_(NDICP, measured)_(6)

Based on this, a corrected pre-cecal digestible Lys (pcdLys_(corrected)_) was calculated as:pcdLys_(corrected)_ = 0.9 × Lys_(NDSCP, measured)_(7)

The metabolizable energy (ME) was calculated according to the GfE equation [1]:ME (MJ/kg DM) = −3.54 + 0.0129 × CP + 0.0420 × CL − 0.0019 × CX + 0.0185 × NFE^1^(8)
^1^NFE (g kg^−1^ DM) = DM − (CP + CL + CF + CA)(9)

### 2.4. Statistical Analysis

Data analysis was performed using the statistical software program STATISTICA 12.0 (StatSoft GmbH, Hamburg, Germany). All data were tested for normal distribution using the Shapiro–Wilk test. 

#### 2.4.1. Feedstuffs

For pcDCP in feedstuffs, linear regression analysis was applied to correlate aNDFom with NDSCP. Analysis of variance (ANOVA) with Bonferroni correction was performed to compare the estimated pcdCP values of the feedstuff groups. 

#### 2.4.2. Feeding Trial

For ppr plasma Lys levels, repeated measures ANOVA was performed, factoring the effects of the diet and time post-prandially. Fisher’s least significant difference (LSD) *post hoc* test was applied in cases of significant time or diet effects. Horses that refused the test meal or had leftovers were excluded from the respective statistics. Finally, 21 test meals were assessed for the statistics: CTRL (*n =* 7), CTRL + synSAA (*n =* 4), and CTRL + SBM (*n =* 8). Only two horses had no leftovers for LUC; therefore, no statistical analysis was performed for LUC. Plasma could not be sampled before feeding the LUC for horse 5 (Table A4). The Mann–Whitney U test was used to assess differences in fasting Lys levels. To compare the mean changes in plasma Lys, the area under the curve (*AUC*) was calculated as follows:(10)$$AUC=\overset{x=1}{\underset{n}{\sum }}tx$$
where *tx* is the period in minutes between the blood sampling time points *t* (*x* − 1) and *t* (*x*). The Mann–Whitney U test was performed to determine the effect of the diet. Statistical significance was set at *p* < 0.05.

## 3. Results

### 3.1. Analytical Protein Evaluation

#### 3.1.1. Protein Fractions of the Feedstuff Groups

The results of the 71 feedstuff samples are shown in Table 1. The percentage of the NDICP fraction in the feed did not differ significantly between the groups. The NDSCP value was the highest for legumes and seeds compared to the other groups (*p* < 0.01). No significant differences in NDSCP levels were observed between the remaining feedstuff groups (*p* > 0.05). 

No significant differences in mean NDICP content were found between the feedstuff groups (*p* > 0.05). The highest amount of NDSCP was found in legumes and seeds, with a median value of 260 g kg^−1^ DM (Table 1). Compound feeds and the group “Others” also contained a high proportion of potentially pre-cecally digestible protein with a median NDSCP content of 115–116 g kg^−1^ DM. Most of these feeds were compound feeds that contained legumes and seeds as single components in varying amounts. Grass contained a soluble protein fraction with a median value of 112 g kg^−1^ DM (Table 1). Grains and roughages had comparable median CP and NDSCP contents (Table 1).

According to the calculation of the pcDCP, the following ranking of the feedstuff groups was determined (in descending order): legumes and seeds (76.0%), compound feeds (69.4%), grains (64.6%), others (62.3%), grass (59.3%), and roughages (55.6%). The pcDCP of legumes and seeds was higher than that of roughages (*p* < 0.01) and grass (*p* < 0.05). The compound feeds had a higher estimated pcDCP ranking than the roughage group (*p* < 0.05). There were no significant differences in the estimated pcDCP between the other groups (*p* > 0.05, Table 1).

#### 3.1.2. Regression Analysis between Selected Parameters of Protein Evaluation

The CP content of the feeds (*N =* 71) was correlated with the estimated pcdCP level (*p* < 0.01). The linear relationship between CP and estimated pcdCP can be described by the following equation:pcdCP (g kg^−1^ DM) = −32.36 + 0.860 × CP (g kg^−1^ DM) (r = 0.967; *p* < 0.01)(11)

Furthermore, aNDFom was negatively correlated with the percentage of pcDCP. The linear correlation can be described by the following equation:pcDCP (%) = 77.75 − 0.039 × aNDFom (g kg^−1^ DM) (r = −0.573; *p* < 0.01)(12)

Other crude nutrients such as CF were not significantly (*p* > 0.05) correlated with estimated pcdCP or pcDCP.

### 3.2. Feeding Trial

#### 3.2.1. Lys in the Protein Fractions of the Test Diets

Lys was not equally distributed over the protein fractions NDSCP and NDICP (Figure 1). The relative Lys content of the pre-cecally undegradable NDICP was highest for SBM (13.9%), whereas synAA had the lowest relative Lys content in the NDICP (3.19%). A Lys content of NDICP of 7.98% was observed for LUC.

Consequently, the calculated pcdLys needed to be corrected using the measured Lys content of NDSCP/NDICP for the estimation of pcdLys_(corrected)_ (Table 2). Correcting pcdLys for synAA and SBM resulted in a difference of about 1.6–4.5%. For LUC, pcDLys decreased from 58.2% to 36.5%.

#### 3.2.2. Clinical Health Status and BW

During the feeding period, none of the horses showed clinical signs of disease or discomfort. The mean BW (±SD) remained constant, at 622 ± 27 kg at the beginning and 617 ± 35 kg at the end of the trial.

#### 3.2.3. Feed Intake

The mean (±SD) meal size and the feed intake time per meal (min) are presented in Table 3. The feed intake per meal was not significant between the different diets (*p* < 0.05).

#### 3.2.4. Lysine Intake

The correction of the pcdLys resulted in a final pcdLys_(corrected)_ amount of 0.74 g/100 kg BW for the CTRL diet, 2.37 g/100 kg BW for CTRL + SBM, and 1.10 g/100 kg BW for LUC (Table 4). The highest amount of estimated pcdLys_(corrected)_ was found in the CTRL + synAA diet at 2.62 g/100 kg BW.

#### 3.2.5. Fasting Levels of Plasma Lys

The mean fasting plasma Lys concentrations (*N =* 31) of the feeding groups were 63.1 ± 32.4 µmol/L (CTRL, *n =* 8); 67.8 ± 27.7 µmol/L (CTRL + synAA, *n* = 8); 57.1 ± 16.6 µmol/L (CTRL + SBM, *n =* 8); and 70.2 ± 24.5 µmol/L (LUC, *n =* 7). No significant differences in fasting levels among the diets were observed (*p* > 0.05, Table A4).

#### 3.2.6. Post-Prandial Changes in Plasma Lys

The post-prandial plasma Lys concentrations increased within 30 to 120 min for CTRL + synAA and CTRL + SBM (Table 5).

After feeding CTRL, plasma Lys increased within 30 min ppr to reach a maximum mean (±SD) peak of 65.7 ± 31.8 µmol/L. At 90 min after feeding, the plasma Lys concentration returned to below the pre-feeding level and remained below this level for 7 h ppr.

The plasma Lys levels increased to a mean (±SD) maximum of 118 ± 55 µmol/L 60 min after CTRL + synAA intake (*p* < 0.05). Subsequently, plasma Lys decreased over a 7 h period without returning to basal concentrations. 

In comparison, for feeding with the CTRL + synAA diet, the mean plasma Lys concentrations increased to a similar extent as for feeding with CTRL + SBM (*p* < 0.05). The plasma Lys levels were not significant between the different diets (diet *p* > 0.05).

Horses 6 and 8 showed maximum plasma Lys levels at 120 min ppr after feeding with LUC (Lys_Max_ horse 6, 68.9 µmol/L; Lys_Max_ horse 8, 117 µmol/L). After reaching the maximum level, the plasma Lys content returned to baseline levels or fell below baseline concentrations.

The mean *AUC* of plasma Lys was the lowest for CTRL (Table 6). In comparison to CTRL, the *AUC* of plasma Lys was significantly higher for CTRL + synAA (*p* < 0.05) and CTRL + SBM (*p* < 0.01).

The diets were ranked in descending order according to their mean (±SD) relative increase from the fasting plasma Lys level up to the Lys peak as follows (Table A4): CTRL + synAA (110 ± 37.6%), CTRL + SBM (92.3 ± 47.7%), LUC (75 ± 33.6%), and CTRL (14.4 ± 13.5%).

## 4. Discussion

The present study was conducted to analyze typical horse feeds using the GfE protein evaluation system [1]. In addition to the chemical approach, we investigated post-prandial changes in plasma Lys to assess the availability of Lys in different feeds for horses. We hypothesized that a higher proportion of estimated pcdLys content in a diet would be reflected by respective post-prandial increases in plasma Lys.

The feeds covered the spectrum of typical feeds for horses, including the most important feed groups, such as roughage, cereals, legumes, high-fat seeds, and complementary feeds for horses with specific requirements, such as breeding, sport, and special breeds.

The hypothesis that a higher amount of CP is related to a higher pcdCP content in the analyzed feeds was confirmed. However, our data also emphasize that a mean NDICP content of at least 3–5% can be assumed in horse feeds from all feedstuff groups. For protein evaluation, the differences in NDF content and the distribution of AAs, such as Lys, must be considered. Similar results have been shown for protein-rich complementary feeds for breeding horses [16].

In contrast to our expectations, our study did not confirm that a higher CF content can predict a lower pre-cecal digestibility. These findings are probably related to the composition of CF. CF consists of cell wall components such as lignin and, partially, cellulose, but does not contain hemicellulose. Therefore, the analysis of NDF, which includes hemicellulose, in equine feeds and the determination of the nitrogen fractions in the NDF seems to be more suitable for estimating the content of pre-cecally digestible protein. This assessment was confirmed by the close negative relationship between NDFom and the percentage of pcDCP.

Because of the relationship between estimated pcdCP and NDF in our study, the equation based on CP and NDF content seemed adequate for a rough estimation of the pcdCP content.

The evaluation system of the GfE [1] estimates the pcdCP based on an analytical determination of CP and NDICP. For the estimation of pre-cecal AA content, an equal AA profile in the NDICP and NDSCP fractions was assumed. However, to date, only three studies [13,14,17] have confirmed an equal distribution of amino acids in different protein fractions in roughages. Although our data were limited by the small sample size, our results might indicate an unequal distribution of Lys over the protein fractions, at least for LUC, SBM, CTRL, and synAA. Accordingly, the potential pre-cecally digestible Lys content was corrected for the feedstuffs used in the feeding trial. For LUC, Lys seemed to be located predominantly in NDICP rather than NDSCP. However, these findings might be influenced by the maturity of the lucerne during harvesting. In this context, increasing maturity of lucerne might lead to a shift in protein degradability from soluble NDSCP to the insoluble NDICP fraction [31]. With increasing maturity, the proportion of NDICP content shifts in favor of the lignin-bound nitrogen content (ADICP), representing a protein fraction that is not degradable in the equine hindgut [32]. It is unclear whether the AA profiles of NDICP and ADICP remain constant or whether they differ. Consequently, the increasing maturity of lucerne could result in decreasing Lys availability as more Lys is bound in the ADICP insoluble fraction. The mean variation in NDF degradability of mature lucerne ranges from 24–41% in horses [33,34,35,36]. However, these data were obtained according to the overall apparent digestibility of the NDF throughout the gastrointestinal tract. There is a knowledge gap regarding the extent of small intestine NDF degradability. Additionally, studies describing the impact of the chewing process and microbial NDF degradability of feedstuffs in equines, for example, lucerne, are missing. Although no data on NDF fermentation in the small intestine of horses are available, in pigs, pre-cecal NDF fermentation of up to 17% has been described in the small intestine [37]. Thus, it can be speculated that microbial pre-cecal NDF fermentation might also play a role in horses. Studies in fistulated horses have reported bacterial cellulolytic activity in the small intestine, ranging from 1.4–32 × 10^8^ colony forming units g^−1^ chyme [38], suggesting that the contribution of microbiota to pre-cecal NDF fermentation in horses is at least similar to that in pigs [39]. Accordingly, the unequal distribution of Lys over the protein fractions in lucerne may lead to an underestimation of pcdLys. Further research is needed to elucidate the utilization of AA in the pre-cecal fermentation of NDF.

The GfE postulates that estimated pcdCP and pcdLys from feeds are only degraded in the small intestine, and that they subsequently enter the bloodstream via absorption from the small intestine. The increase in peak plasma Lys within 30 to 120 min ppr is likely to mirror the absorption from the small intestine. Our results (plasma Lys peak at 60 min ppr for CTRL + SBM and CTRL + synAA) are in accordance with previously obtained results [23,40,41,42]. 

According to other studies, the diet composition and feeding schedule on ppr changes in plasma Lys and other AAs in equines have already been reported [15,22,23,24,37,40,41,42,43,44,45,46,47,48,49].

For all test diets, the relative increases in fasting plasma Lys levels to the mean maximum values were similar to those plasma Lys levels after feeding soybean meal, fish meal, or linseed meal [24]. In contrast, a relative increase of >150% was reported when hay was fed before concentrate [22,44]. Furthermore, factors such as age, fasting period before the feeding of the test diet, and protein source have been reported to influence ppr Lys changes [15,46]. To the best of our knowledge, this is the first study investigating the effect of the solubility of the CP fractions and the distribution of Lys over the protein fractions on the ppr changes in plasma Lys. The significantly higher *AUC* of CTRL + SBM and CTRL + synAA compared to CTRL confirmed our assumption that a higher Lys content, especially when Lys is provided as a free amino acid or by a highly digestible Lys source, is reflected in the ppr changes of Lys in the plasma. Nevertheless, different factors, such as pre-cecal NDF fermentation and the chewing process, should be included in the protein evaluation of feedstuffs. The similar pcdLys_(corrected)_ values of LUC and CTRL contrasted with the ppr changes in plasma Lys.

However, it must be emphasized that in protein providers such as SBM, the unequal distribution of Lys over the protein fractions has little impact on ration formulation because of the low proportion of NDICP in relation to the total protein. Consequently, the major proportions of CP and AA content in protein providers (e.g., SBM or synAA) can be expected to be pre-cecally digestible.

## 5. Conclusions

As expected, using the new evaluation system, protein-rich feedstuffs such as soybean by-products, peas, and brewer’s yeast were classified by a high pcDCP. We recommend chemical analyses of feedstuffs be made mandatory for the estimation of pcdCP. 

Further investigations are necessary to define the impact of chewing and microbial activity on pre-cecal NDF fermentation, as these factors influence the availability of AA.

## Figures and Tables

**Figure 1 animals-13-02624-f001:**
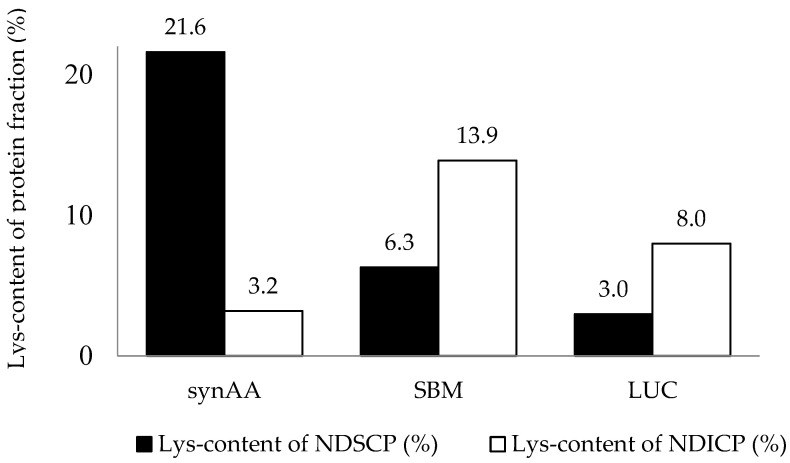
Lysine content in the protein fractions NDSCP and NDICP (expressed in %) of three test feeds (synAA, SBM, and LUC). NDICP: neutral detergent insoluble crude protein; NDSCP: neutral detergent soluble crude protein; synAA: supplement with synthetic amino acids; SMB: soybean meal (solvent extracted); LUC: lucerne pellets.

**Table 1 animals-13-02624-t001:** Chemical composition of six feedstuff groups (g kg^−1^ DM) and parameters of protein evaluation according to the German Society of Nutrition Physiology.

Groups	Roughages	Grass	Legumes and Seeds	Compound Feeds	Grain	Others
	(*n =* 19)	(*n =* 11)	(*n =* 9)	(*n =* 11)	(*n =* 11)	(*n =* 10)
CA	94.8 (37.3–131)	89.1 (75.9–145)	50.0 (32.5–89.3)	66.7 (45.3–120)	26.3 (7.0–81.7)	60.4 (12.3–130)
CF	307 (189–405)	258 (176–329)	71.8 (37.0–413)	80.8 (49.6–115)	30.0 (0–144)	88.6 (7.8–254)
CL	19.1 (8.8–31.2)	27.3 (22.8–145)	144 (12.4–459)	44.7 (36.6–91.9)	24.8 (9.2–167)	48.5 (6.3–306)
NFE	477 (382–664)	447 (256–470)	324 (116–648)	624 (496–683)	799 (455–858)	580 (489–665)
aNDFom	552 (389–749)	539 (456–616)	235 (74.1–641)	223 (168–321)	209 (47.7–311)	272 (21.8–547)
ADFom	354 (202–494)	279 (210–410)	126 (73.6–447)	83.0 (52.9–174)	87.4 (0–164)	145 (7.7–291)
ADL	60.9 (21.2–103)	48.4 (20.5–194)	32.6 (0–137)	21.2 (3.8–66.5)	26.6 (0–52.5)	24.1 (0–99.9)
CP	121 (32.9–184)	178 (107–263)	359 (112–485)	160 (107–265)	119 (85.5–152)	153 (18.1–350)
NDICP	42.6 (18.0–79.9)	47.1 (39.3–108)	31.3 (21.3–110)	37.2 (20.4–47.4)	20.9 (14.8–69.2)	38.6 (6.2–142)
NDSCP ^1^	68.7 (15.0–137)	112 (60.2–162)	260 (69.3–463)	115 (86.6–217)	84.8 (38.8–132)	116 (11.9–335)
pcdCP	61.8 (13.5–123)	101 (54.2–146)	234 (62.3–417)	104 (77.9–196)	76.3 (34.9–119)	105 (10.7–302)
pcDCP	58.4 (38.8–67.4)	58.3 (50.5–70.0)	79.3 (55.8–86.0)	72.0 (60.2–76.5)	71.5 (32.3–78.6)	61.2 (37.8–86.4)

Values expressed as median with minimum and maximum in round brackets; DM, dry matter; CA, crude ash; CF, crude fiber; CL, crude lipid; NFE, nitrogen-free extract; aNDFom, neutral detergent fiber treated with a heat-stable amylase and expressed exclusive of residual ash; ADFom, acid detergent fiber, expressed exclusive of residual ash; ADL, acid detergent lignin; CP, crude protein; NDICP, neutral detergent insoluble crude protein; NDSCP, neutral detergent soluble crude protein; pcdCP, estimated pre-cecal digestible crude protein; pcDCP, pre-cecal digestibility of crude protein; ^1^ calculated according to the German Society of Nutrition Physiology and Zeyner et al. [26].

**Table 2 animals-13-02624-t002:** Lysine evaluation based on the estimated and analyzed NDSCP/NDICP lysine contents of three test feeds (synAA, SBM, and LUC).

Parameter	synAA	SBM	LUC
Estimated ^1^			
pcDLys ^1^ (%)	95.0	85.8	58.2
Analyzed ^2^			
pcDLys_(corrected)_ ^2^ (%)	93.4	81.3	36.5

^1^ Calculated according to the GfE [1] and Zeyner et al. [26]; ^2^ calculated according to the GfE [1] and Zeyner et al. [26] using HPLC to analyze Lys contents of the protein fractions NDSCP and NDICP. NDICP: neutral detergent insoluble crude protein; NDSCP: neutral detergent soluble crude protein; synAA: supplement with synthetic amino acids; SMB: soybean meal (solvent-extracted); LUC, lucerne pellets; pcDLys, pre-cecal digestibility of lysine.

**Table 3 animals-13-02624-t003:** Feed intake time and meal size with the CTRL diet (standardized to 1.02 g/Lys intake per 100 kg body weight) and the three test diets (CTRL + synAA, CTRL + SBM, and LUC), standardized to 3.02 g/Lys intake per 100 kg body weight fed to horses (expressed as mean ± standard deviation).

Test Meals	CTRL (*n =* 7)	CTRL + synAA (*n =* 4)	CTRL + SBM (*n =* 8)	LUC (*n =* 2)	
				Horse 6	Horse 8
Meal size (kg FM) ^1^	1.90 ± 0.08	2.07 ± 0.08	2.31 ± 0.12	2.57	2.75
Feed intake time per meal (min)	19.3 ± 9.0	18 ± 6.0	22.0 ± 5.5	86	55

^1^ FM: Fresh matter; CTRL: muesli for sport horses; synAA: supplement with synthetic amino acids; SMB: soybean meal (solvent-extracted); LUC: lucerne pellets.

**Table 4 animals-13-02624-t004:** Intake of dry matter, crude protein, pcdCP and pcdLys(g/100 kg BW) and pcDCP evaluation (%) of a commercial muesli for sport horses (CTRL diet; standardized to 1.02 g Lys intake per 100 kg body weight) and three test diets (CTRL + synAA, CTRL + SBM, and LUC; standardized to 3.02 g Lys intake per 100 kg body weight) fed to horses.

Test Diet	CTRL	CTRL + synAA	CTRL + SBM	LUC
DM intake	270	300	340	420
CP intake	29.3	38.9	61.8	63.6
pcdCP ^1^ intake	21.3	29.6	47.0	37.1
pcDCP of the diet (%) ^1^	73	76	76	58
Total Lys content of the diet	1.02	3.03	3.02	3.02
Synthetic Lys content	n.a.	1.89	n.a.	n.a.
pcdLys ^1^ intake	0.74	2.66	2.46	1.76
pcdLys_(corrected)_ ^2^ intake	0.74	2.62	2.37	1.10

^1^ Calculated according to the GfE [1] and Zeyner et al. [26]; ^2^ calculated according to the GfE [1] and Zeyner et al. [26] using HPLC-analyzed Lys contents of the protein fractions NDSCP and NDICP; n.a.: not applicable; synAA: supplement with synthetic amino acids; SMB: soybean meal (solvent extracted); LUC: lucerne pellets; pcdCP: pre-cecal digestible crude protein; pcDCP: pre-cecal digestibility of protein; pcdLys: pre-cecal digestible lysine; pcDLys: pre-cecal digestibility of lysine; pcdLys_(corrected)_: corrected pre-cecal digestible lysine.

**Table 5 animals-13-02624-t005:** Post-prandial plasma lysine concentrations (µmol/L) of horses fed a commercial muesli for sport horses (CTRL diet, standardized to 1.02 g Lys intake per 100 kg body weight) and three test diets (CTRL + synAA, CTRL + SBM, and LUC; standardized to 3.02 g Lys intake per 100 kg body weight) fed to horses.

Time ppr (min)	CTRL (*n =* 7)		CTRL + synAA (*n =* 4)		CTRL + SBM (*n =* 8)		LUC ^1^ (*n =* 2)	
	Mean	SD	Mean	SD	Mean	SD	Horse 6	Horse 8
0	64.3 ^a^	37.2	59.7 ^a^	16.1	57.1 ^a^	17.7	49.1	85.8
30	65.7 ^a^	31.8	114 ^b^	50.9	84.3 ^a^	32.6	67.5	143
60	65.2 ^a^	29.9	118 ^b^	55.0	106 ^b^	44.3	74.2	134
90	59.2 ^a^	23.2	107 ^a^	53.0	101 ^b^	42.7	60.1	159
120	51.2 ^a^	24.9	95.2 ^a^	46.1	98.7 ^b^	46.2	68.9	171
150	47.2 ^a^	27.0	92.1 ^a^	30.3	84.8 ^a^	38.5	57.4	169
180	48.9 ^a^	25.3	93.2 ^a^	44.6	88.6 ^a^	32.0	61.8	155
210	46.1 ^a^	20.9	94.8 ^a^	56.0	90.8 ^a^	41.3	49.6	132
240	40.3 ^a^	19.5	87.3 ^a^	55.4	91.0 ^a^	41.1	66.1	129
300	39.2 ^a^	20.8	91.4 ^a^	57.0	83.4 ^a^	35.9	71.6	69.6
360	41.5 ^a^	19.4	99.5 ^a^	55.0	78.1 ^a^	33.8	63.2	60.4
420	38.4 ^a^	12.6	99.5 ^a^	45.8	73.8 ^a^	30.2	57.1	79.6
480	50.5 ^a^	30.5	86.6 ^a^	40.3	69.9 ^a^	37.3	50.9	73.4

^ab^ Different superscript letters indicate significant (*p* < 0.05) differences within the columns compared to time point 0; synAA: supplement with synthetic amino acids; SMB: soybean meal (solvent extracted); LUC: lucerne pellets; ppr: post-prandial. ^1^ LUC is expressed as the individual plasma Lys concentrations of horse 6 and 8.

**Table 6 animals-13-02624-t006:** Area under the curve (*AUC*) of plasma lysine for a commercial muesli for sport horses (CTRL diet; standardized to 1.02 g Lys intake per 100 kg body weight) and three test diets (CTRL + synAA, CTRL + SBM, and LUC; standardized to 3.02 g Lys intake per 100 kg body weight) fed to horses.

Parameter	CTRL ^1^ (*n =* 7)	CTRL + synAA ^1^ (*n =* 4)	CTRL + SBM ^1^ (*n =* 8)	LUC ^2^ (*n =* 2)	
				Horse 6	Horse 8
*AUC* Lys (µmol × min/L)	22,944 ^a^ ± 9940	46,262 ^b^ ± 18,858	40,768 ^b^ ± 15,399	29,938	53,680

^1^ *AUC* is expressed as mean ± SD; ^2^
*AUC* is expressed as individual values of horse 6 and 8. ^ab^ Means in the same row with different superscripts are significantly different at *p* < 0.05. synAA: supplement with synthetic amino acids; SMB: soybean meal (solvent extracted); LUC: lucerne pellets.

## Data Availability

The datasets used and analyzed during the current study are available from the corresponding author upon reasonable request.

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
