# Peer review of "Protein Evaluation of Feedstuffs for Horses"

_animals, 2023, doi:10.3390/ani13162624_

Round 1

Reviewer 1 Report

See attached file

Sentence structure and language can be improved. 

Reviewer 2 Report

The manuscript contains an up-to-date topic and delivers relevant information regarding protein evaluation in feedstuffs for horses. In the present state, it is partly hard to understand. Some revisions should be made regarding language (decide for British or American English; use the term “protein evaluation system” instead of “protein system”, check if you need to use an adjective or an adverb – for example “pre-caecal digestibility” but “pre-caecally digestible”, be consistent with the names of the trials – “feeding trial” or “in vivo trial” and “analytical protein evaluation” or ?) and sorting order of information. The use of appendix-tables is not comfortable for the reader. At some points information is redundant or absent. As indicated in the author guidelines, please use SI units and the associated style (g kg-1 instead of g/kg).

Simple summary

12 protein evaluation system

18ff not congruent with conclusion

Abstract

Only the feeding trial is described – the evaluation of the 71 feedstuffs is not described

23 pre-ceacally digestible

29 abbreviation/explanation for AUC is lacking

Keywords

NDF-bound crude protein

Introduction

38 of horses.

42 there is data regarding absorption of AA from the hindgut. Maybe: “protein breakdown in the hindgut is negligible

43 hard to understand. Please reframe. For example: “Microbial digestion occurs not exclusively but significantly in the large intestine. This necessitates the analysis of the NDF-bound proportion of crude protein, which is mainly microbially digested.”

58 delete “within these two protein fractions”

59 Like for the protein, also for the AA it is assumed that 90 % of the NDSCP- bound fraction is pre-caecally digestible (pcdAA).

62ff Please rephrase the next two paragraphs. Decide, which information might be better placed in Materials and Methods. Add your hypothesis (In the discussion, you address hypothesis, which are not mentioned before, like correlations with CF,…).

Materials and Methods

Please sort your information more clearly.

76ff give information, why these feedstuffs were chosen and why senior muesli was sorted in “Others”

81ff The table should be used in the results. Why have legumes and seeds a high standard deviation?

97 ground to pass a 1 mm sieve

98 were the samples, which were dried at 103°C used for Weende Analyses? Please describe drying and storage of the samples

113 Appendix tables should be put in the text – A1 can be combined with A2 Table 1 and put in the results section.

138 Are the calculations for both experiments? If yes, combine them with the section “statistical analysis”. If not, it would be 2.1.4 Calculations

158 please divide the paragraphs: start a new paragraph with “The pre-caecal digestibility…”

168 please describe, why “corrected” is x0.9

181 2.2 Information on location and trial number (lines 203 and 210). The following paragraphs should be 2.2.1 Animals, 2.2.2 Feedstuffs and diets, 2.2.3 Experimental design, 2.2.4 Blood sampling day, 2.2.5 Sample preparation…, 2.3 Statistical analysis, 2.3.1 Chemical protein evaluation, 2.3.2 Feeding trial

Tables A3 and A4 should be combined and put in 2.2.2

195 only 73% of the diet

202 include age of the geldings in discussion

237-239 should be the last sentence of the paragraph

259 please make clear, which statistics were used for which study and which test was used for which feedstuff/horse?

282ff Table 1 (combined table 1, A1, A2) instead of Figure. Figure is not comprehensive.

312ff why did you use a correlation and not a regression?

 325ff Did you not analyse Lys in the NDSCP and NDICP of the 71 feedstuffs? It is confusing to find results of the feeding trial feedstuffs here. Please make the description clear and concise. Please delete the percentages in the text (they are in the figure) and in general decide on a number of decimal places.

338ff this paragraph is difficult to understand. Please be concise in the use of the terms “corrected” “measured” “estimated” and calculated” (again in l 371). The information of table 2 can be given in one sentence in the text, making the table dispensable.

355ff Please do not repeat numbers in the text, which are already in the table. Please combine tables 3 and 4.

377 Title of table ”protein and Lysine evaluation” is not a component – please specify for the table

387ff The cited data is not to be found in Table A5 (38 and 91 μmol/L). You can either use the table in the text: “No significant differences in fasting levels among the diets were observed (Table x).” or include its information: “The mean fasting plasma Lys-concentration of the feeding groups was 61.1 ± 32.4, 67.8 ± 27.7, 57.1 ± 16.6, 70.2 ± 24.5 for CTRL (n = 8), CTRL+synAA (n = 8), CTRL+SBM (n = 8) and LUC (n = 7), respectively.”

394 Table 5. Post-prandial plasma Lysine concentrations (μmol/L) of horses fed a commercial…

421 Table A6 is redundant

Discussion

433 Please clarify your aim. Did you want to analyse feedstuffs or to evaluate the system?

442 and 448 where can I find these hypothesis?

456 do you consider r=-0.57 as adequate?

457f comparison with your equation?

501 hindgut?

548 Author contribution statement not included

Please check spelling/formatting of your references

Round 2

Reviewer 2 Report

All in all, the manuscript is improved. The topic is interesting and the study is well conducted.

There are many volatility errors in the revised text. That does not make a good impression. Furthermore, some colloquial expressions are used (see below, f. e. lines 152, 395, 441,...). Check for those errors and for consistency again. The following comments should be taken into account.

Please note that "lysine" should be written in lower case letters even if your abbreviation "Lys" begins with a capital letter. Again: please check, if you need to use the adverb (pre-caecally) or the adjective (pre-caecal). It seems that you used CTRL+F to search and replace some phrases resulting in double full stops und such things. please check.

25 feedingtrial – feeding trial

34 italics

47 pre-caecally digestible (twice)

48 two full stops

53 pre-caecally

56 pre-caecally

58 pre-caecally

66 hypothesized american English – hypothesized

68f is not a comprehensive sentence

101 1 mm sieve – the full stop is not there

102 ,.

102 After oven drying, …

103 desiccator (not exicator)

122 You wrote that lysine was not analysed in the 71 samples. In that case, 2.1.3 should be 2.2.3. If lysine was analysed in the feedstuffs but not in the NDF fraction, please specify and include in table 1.

146 2.2 twice

149 I, ii, 3, iv – please change 3 to iii

149ff information is redundant (line 199ff) It would be more comprehensive, if you gave the information only in the next paragraphs. In this paragraph, only the following information is necessary: “A feeding trial with eight horses was conducted at Leipzig University. The project was approved by the Ethics Committee for Animal Rights Protection of the Leipzig District Government (TVV 18/15), in accordance with the German legislation 168 for animal rights and welfare.”

152 contained 1.02g/lys intake? It contained an amount per mass or it led to an intake of an amount per body mass?

167ff information in first paragraph (2.2)?

223 remove a blank space

328 [.30] – [30]

333 potential pre-caecally

366 pre-caecally

391 The BW remained constant with a BW (mean ± SD) of 622 ± 27.4 kg at the beginning and a BW (mean ± SD) of 617 ± 35.0 kg at the end of the trial. à The mean BW (± SD) remained constant with 622 ± 27.4 kg at the beginning and 617 ± 35.0 kg at the end of the trial.

395 two full stops

395 Feed intake time per meal did not differ significantly between the diets (p < 0.05). (Time can not be significant. Only a difference can be significant or not.)

411 abbreviations do not fit at the points, where they are included.

411 lysine

422 The mean (±SD)?

429 lysine

441 CTRL + synAA increased the maximum plasma level to a mean of… (not the diet increased but the mean plasma lysine level after the intake of the diet)

442 two full stops

443 two full stops

444f In comparison, mean plasma Lys concentrations increased to a similar extent when CTRL + SBM were fed. (compared to which diet?)

445 Not the levels were not significant, but the difference between the levels! Please rephrase.

452 control diet = CTRL (consistency again)

458 lysine (Lys)

482 In contrast to our expectations, our study did not confirm that a higher CF content may predict a lower precaecal digestibility.

485ff Therefore, the analysis of NDF, which includes hemicellulose, in equine feeds and the determination of the nitrogen fractions in the NDF seems to be more suitable for estimating the content of pre-caecally digestible protein.

491 …a rough estimation of the pcdCP content. (predicting redundant)

490ff please discuss the difference between their equations and yours.

494f On which basis are those equations made, if there are no analyses?

500;.

502 ,.

502 pre-caecally

515ff Additionally, studies are missing to describe the impact of the chewing process and microbial NDF degradability of feedstuffs in equines, for example, lucerne. à Additionally, studies describing the impact of the chewing process and microbial NDF degradability of feedstuffs in equines, for example Lucerne, are missing.

522 /g à g-1

References: you exchanged commas between the authors’ names with full stops. Please undo.

624 two full stops

634 Luther University: Martin. Halle. 1995?

660 Proteinwerte, vonam

I did not check, whether the references in the reference list and the text are congruent. Please be sure they are.

see in comments and suggestions

Author Response

Dear editors

We would like to thank the Associate Editor and the Reviewer to have the chance to revise the manuscript a second time.

The comments are attachde in a seperate file.

Kind regards

Ingrid Vervuert
